# Certified Robustness against Sparse Adversarial Perturbations via Data Localization

**Ambar Pal**  AMBAR@JHU.EDU
*Department of Computer Science &*
*Mathematical Institute for Data Science*
*Johns Hopkins University*
*Baltimore, MD 21218, USA*

**René Vidal**  VIDALR@UPENN.EDU
*Department of Electrical and Systems Engineering &*
*Center for Innovation in Data Engineering and Science*
*University of Pennsylvania*
*Philadelphia, PA 19104, USA*

**Jeremias Sulam**  JSULAM1@JHU.EDU
*Department of Biomedical Engineering &*
*Mathematical Institute for Data Science*
*Johns Hopkins University*
*Baltimore, MD 21218, USA*

**Reviewed on OpenReview:** *https://openreview.net/forum?id=17Ld3davzF*

## Abstract

Recent work in adversarial robustness suggests that natural data distributions are localized, *i.e.*, they place high probability in small volume regions of the input space, and that this property can be utilized for designing classifiers with improved robustness guarantees for $\ell_2$-bounded perturbations. Yet, it is still unclear if this observation holds true for more general metrics. In this work, we extend this theory to $\ell_0$-bounded adversarial perturbations, where the attacker can modify a few pixels of the image but is unrestricted in the magnitude of perturbation, and we show necessary and sufficient conditions for the existence of $\ell_0$-robust classifiers. Theoretical certification approaches in this regime essentially employ voting over a large ensemble of classifiers. Such procedures are combinatorial and expensive or require complicated certification techniques. In contrast, a simple classifier emerges from our theory, dubbed BOX-NN, which naturally incorporates the geometry of the problem and improves upon the current state-of-the-art in certified robustness against sparse attacks for the MNIST and Fashion-MNIST datasets.

## 1 Introduction

It is by now well known that adversarial attacks affect Machine Learning (ML) systems that can potentially be used for security sensitive applications. However, despite significant efforts on robustifying ML models against adversarial attacks, it has been observed that their performance on most tasks under adversarial perturbation is not close to human levels. This motivated researchers to obtain theoretical impossiblity results for adversarial robustness Shafahi et al. (2018); Dohmatob (2019); Dai & Gifford (2022), which state that for general data distributions, no robust classifier exists against adversarial perturbations, even when the adversary is limited to making small $\ell_p$-norm-bounded perturbations. However, such results are seemingly in conflict with the fact that humans can classify most natural images quite well under small $\ell_p$-norm-bounded perturbations. Even more, there is a rich literature on certified robustness, *e.g.*, Zhang et al. (2018); Cohen et al. (2019); Pal & Vidal (2020); Fischer et al. (2020); Jeong & Shin (2020); Jia et al. (2022); Pfrommer

et al. (2023); Salman et al. (2022); Eiras et al. (2022); Pal & Sulam (2023), where the goal is to obtain and analyze methods with provable guarantees on their robustness under adversarial attacks.

Pal et al. (2023) recently provided a solution to this apparent conflict, noting that existing impossibility results become vacuous when the data distribution is such that a large probability mass is concentrated on very small volume in the input space, a property they call $(C, \epsilon, \delta)$-concentration. This characterization implies that at least $1 - \delta$ probability mass is found in a region of volume at most $Ce^{-n\epsilon}$ for small $\delta \approx 0$ and large $\epsilon$. As an example, this property dictates that sampling a random $224 \times 224$ dimensional image is extremely likely to *not* be a natural image. This property is intuitively satisfied for natural datasets like ImageNet, and Pal et al. (2023) formally show that whenever a classifier robust against small $\ell_2$-bounded attacks exists for a data distribution (e.g., humans for natural images), this distribution must be concentrated. This shows that indeed, robust classifiers against $\ell_2$ attacks can be obtained for natural image distributions, and there is no impossibility.

While these results are encouraging, attacks that are bounded in Euclidean norm have nice analytical properties that facilitated the results in Pal et al. (2023). In this work, we seek to understand if similar notions can provide insights on provable defenses against *sparse* adversarial attacks (bounded in their $\ell_0$ distance) where the adversary is limited to modifying a few pixels on the image, but those pixels can be modified in an unbounded fashion. Even though for humans it seems trivial to correctly classify a natural image corrupted in a few pixels, this problem has stood out as a particularly hard task for machine learning models. The difference is extreme: Su et al. (2019) demonstrated that adversarially modifying a *single* pixel leads to large performance degradation of many state of the art image recognition models. Standard ideas for improving robustness, like adversarial training, seem to be empirically ineffective against sparse attacks. Since then, researchers have resorted to enumerating a large number of subsets of the input pixels, and taking a majority vote over the class predicted from each subset, as a means of obtaining classifiers robust to sparse attacks. The resultant methods (Levine & Feizi, 2020b) are expensive, and need probabilistic certificates due to the combinatorial blow-up in the number of subsets needed as the number of attacked pixels increases. Follow-up work by Jia et al. (2022) has employed complicated certification schemes to reduce the slack in these certificates, while still remaining computationally expensive. Most recently, Hammoudeh & Lowd (2023) carefully selected these subsets to speed up the certificate computation. However, none of these existing methods utilizes the geometry of the underlying data distribution highlighted by our results. Departing from this stream of research, we propose a classifier that closely utilizes this underlying geometry to obtain robustness certificates. As a result, we provide a classifier that is lighter and simpler than all existing works, and an associated certification algorithm with $\ell_0$ certificates that are better than prior work.

Our proof techniques extend results in Pal et al. (2023) to sparse adversarial attacks. In practice, one can always project the pixel values to lie in some predefined range, say $[0, 1]$, before classification, so we can consider adversarial perturbations to lie within $[0, 1]^n$ without any loss of generality. In other words, our adversary at power $\epsilon$ is allowed to modify an image from $x$ to $x'$ such that $\|x - x'\|_0 \le \epsilon, \|x'\|_\infty \le 1$. The techniques in Pal et al. (2023) break down under such an adversary, as their first assumption is to restrict attention to adversarial perturbations $v$ such that $x + v$ cannot lie $\epsilon$-close to the boundary of the image domain. In our case, the geometry of the problem is radically different: even a perturbation of size 1 is sufficient to take any image to the boundary of the domain $[0, 1]^n$ (simply perturb any pixel to 1). As a result, although we are motivated by Pal et al. (2023), our theory and certification algorithms are markedly different from those in that work.

In the above setting, we show that whenever there exists a classifier robust to adversarial modification of a few entries in the input, the underlying data distribution places a large mass, i.e., *localizes*, on low-volume subsets of the input space. We further show that the converse holds too, albeit with a strengthening of the localization condition; i.e., we show that when the data distribution localizes on low-volume subsets of the input space, and these subsets are sufficiently separated from one another, then a robust classifier exists. These results suggest that such underlying geometry in natural image distributions should be exploited for constructing classifiers robust against $\ell_0$ attacks. Indeed, we then propose a simple classifier, called Box Nearest Neighbors (Box-NN), that utilizes this underlying geometry by having decision regions that are unions of axis-aligned rectangular boxes in the input space. Such a classifier naturally allows for $\ell_0$ robustness certificates that improve upon prior work for certified defenses in a wide regime.

To summarize, we make the following contributions in this work:

1. In Section 2 we show that if a data-distribution $p$ defining a multi-class classification problem admits a robust classifier whose error is at most $\delta$ under sparse adversarial perturbations to $\epsilon$ pixels, then there is a subset $S$ of volume at most $Ce^{-\epsilon^2/n}$ and a class $k$ such that the class conditional $q_k$ places a large mass $q_k(S) \geq 1 - \delta$ on $S$, i.e., $q_k$ is $(C, \epsilon^2/n, \delta)$-localized. The constant $C$ captures certain geometric properties of the robust classifier, and will be clarified later.

2. In Section 3, we show that a stronger notion of localization, which ensures that the class conditional distributions are sufficiently separated with respect to the $\ell_0$ distance, is sufficient for the existence of a robust classifier. In fact, this result generalizes to any distance $d$, showing the existence of a robust classifier w.r.t. perturbations bounded in distance $d$ whenever the data distribution $p$ is strongly localized with respect to $d$.

3. In Section 4, we propose a classifier certifiably robust against sparse adversarial attacks, called Box-NN, and derive certificates of $\ell_0$ robustness for it. We then provide empirical evaluation on the MNIST and the Fashion-MNIST datasets, and demonstrate that Box-NN obtains state-of-the-art results in certified $\ell_0$ robustness.

## 2   Existence of an $\ell_0$-Robust Classifier implies Localization

We will take our data domain to be $[0,1]^n$, to mimic the standard natural image classification tasks[1], i.e., $\mathcal{X} = \{x \colon \|x\|_\infty \leq 1\}$. We will take our label domain to be $\mathcal{Y} = \{1, 2, \ldots, K\}$, and assume that we have a classification task defined by a joint data distribution $p$ over $\mathcal{X} \times \mathcal{Y}$. The marginal distribution over the classes will be denoted by $p_Y$. The conditional distribution $p_{X|Y=k}$ for each class $k \in \mathcal{Y}$ will be denoted by $q_k$[2]. Further, we will say that the classes are *balanced* when $p_Y(k) = \frac{1}{K}$ for all $k \in \mathcal{Y}$.

In this work, we will study adversarial robustness for non-trivial classifiers that are not constant over the entire domain $\mathcal{X}$. For any such classifier $f \colon \mathcal{X} \to \mathcal{Y}$, we recall the standard definition of robust risk $R_d(f, \epsilon)$ against perturbations bounded in a distance $d$ as

$$R_d(f, \epsilon) = \mathop{\mathbb{P}}_{(x,y)\sim p} \left( \exists \bar{x} \in B_d(x, \epsilon) \text{ such that } f(\bar{x}) \neq y \right).$$

Similarly, we define a classifier $f$ to be $(\epsilon, \delta)$-robust with respect to a distance $d$ if the robust risk against perturbations at a distance bounded by $\epsilon$ is at most $\delta$, i.e., $R_d(f, \epsilon) \leq \delta$.

For the rest of this section, we will assume that $p$ defines a task for which one can obtain a classifier $f$ such that $R_{\ell_0}(f, \epsilon_0) \leq \delta$, where $\epsilon_0$ is a non-negative integer denoting the maximum number of co-ordinates that an adversary can perturb. Given such an $f$, we will show that $p$ should satisfy the special property of localization. In other words, we will obtain a necessary condition for $\ell_0$ robustness. This special property of $(C, \epsilon, \delta)$-localization[3] is similar to Pal et al. (2023, Definition 2.2), with a slight modification:

**Definition 2.1** (Localized Distribution, modification of Pal et al. (2023))**.** A probability distribution $q$ over a domain $\mathcal{X} \subseteq \mathbb{R}^n$ is said to be $(C, \epsilon, \delta)$-localized if there exists a subset $S \subseteq \mathcal{X}$ such that $q(S) \geq 1 - \delta$ but $\mathrm{Vol}(S) \leq C \exp(-\epsilon)$. Here, Vol denotes the standard Lebesgue measure on $\mathbb{R}^n$, and $q(S)$ denotes the measure of $S$ under $q$.

Definition 2.1 is similar to Pal et al. (2023, Definition 2.2) but it removes the explicit dimension of the problem, i.e., $n$, from the volume constraint. This allows one to state the results in Pal et al. (2023), as well as ours, under the same definition. Additionally, we rename the property from *concentration* in Pal et al. (2023) to *localization*, in order to distinguish ourselves from the well known notion of *concentration of measure*. These two notions are related and, before proceeding, we compare them in more detail.

---

[1] Albeit with a scaling – natural images are typically stored with each pixel value in $[0, 255]$.

[2] The conditional density is defined in a standard fashion as $q_k(x) = p(x, k)/p_Y(k)$.

[3] Here, the quantity $C$, similar to $\epsilon$ and $\delta$, is a parameter of localization, and controls the *extent* of localization. $C$ can be determined directly given access to a distribution $p$, or indirectly via Theorem 2.2 given access to a robust classifier.

The notion of *measure concentration* from high dimensional probability theory roughly states that for a given large dimension $n$, "a well behaved function $h$ of the random variables $Z_1, Z_2, \ldots, Z_n$ takes values close to its mean $\mathbb{E}\, h(Z_1, \ldots, Z_n)$ with high probability" (Talagrand, 1996). A popular quantification of this notion states that for a metric space $(\mathcal{X}, d)$ and a probability distribution $q$ over $\mathcal{X}$, the concentration function $\alpha$ defined as

$$\alpha_{q,d}(t) = \sup_{S \subseteq \mathcal{X},\ q(S) \geq 1/2} 1 - q(S^{+t}), \tag{1}$$

decreases "very fast" with $t$, where recall that $S^{+t} = \{x \in \mathcal{X} : d(x, S) \leq t\}$. We typically say that $q$ has the property of measure concentration if there is an exponential decay as $\alpha_{q,d}(t) \sim \exp(-\gamma t)$ for all $t \geq 0$, and some universal constant $\gamma$.

In contrast, the definition of $(C, \epsilon, \delta)$-localization requires the existence of $S \subseteq \mathcal{X}$ such that $q(S) \geq 1 - \delta$ and $\mathrm{Vol}\,(S) \leq C \exp(-\epsilon)$. Concentration and localization are similar in the underlying message: most of the mass in $q$ is concentrated near a small region in space. However, the mathematical formalization is different, as localization does not require a fast enough rate of decay of the measure, and hence does not require an underlying metric on the space $\mathcal{X}$. In order to show that a given distribution $q$ localizes, it is sufficient to provide a single instance of a set $S \subseteq \mathcal{X}$ that satisfies the localization parameters. For our data domain $\mathcal{X} = [0, 1]^n$, we will consider a family of probability distributions given by $q_a = \mathrm{Unif}([0, a]^n)$ for $a \in (0, 1]$, and comment on their localization and measure concentration parameters, to shed light into their similarities and differences.

For any $S \subseteq [0, a]^n \subseteq \mathcal{X}$, we can simplify $1 - \delta \leq q_a(S) = \frac{1}{a^n} \mathrm{Vol}\,(S) \leq \frac{1}{a^n} \exp(-\epsilon)$ to obtain that

$$q_a \text{ is } \left(1, \log\left(\frac{1}{1-\delta}\right) + n \log\left(\frac{1}{a}\right), \delta\right) - \text{localized for any } \delta \in [0, 1].$$

From the above we can see that keeping $\delta, a < 1$ fixed, $q_a$ becomes "more localized" as the dimension $n$ increases. Similarly, keeping $\delta, n$ fixed, $q_a$ becomes more localized as $a$ gets closer to 0. In this sense, the localization parameters depend on the scale of the support of the underlying distribution.

In contrast, as measure concentration depends on an underlying metric, the concentration parameters are independent of the scale of the support when the metric is invariant to scaling. As an example, for $\mathcal{X}$ equipped with the hamming metric, $d_0(x, x') = \|x - x'\|_0$, the concentration function for the distribution $q_a$ can be shown to be

$$\alpha_{q_a, d_0}(t) \leq 2 \exp\left(-\frac{t^2}{n}\right). \tag{2}$$

Armed with the above definition, we will now derive a necessary condition for $\ell_0$-robustness in terms of localization, by using a measure-concentration result w.r.t. the $\ell_0$ distance due to Talagrand (1995).

**Theorem 2.2.** *If there exists a non-trivial $(\epsilon, \delta)$-robust classifier $f$ with respect to the $\ell_0$ distance for a data distribution $p$, then at least one of the class conditionals $q_1, q_2, \ldots, q_K$ must be $(C, \epsilon^2/n, \delta)$–localized according to Definition 2.1. Further, if the classes are balanced, then all the class conditionals are $(C_{\max}, \epsilon^2/n, K\delta)$-localized. Here, $C$ and $C_{\max}$ are constants dependent on $f$.*

*Proof.* We are given a classifier $f$ which is $(\epsilon, \delta)$-robust w.r.t. perturbations bounded in the $\ell_0$ distance. In other words, we have $R_{\ell_0}(f, \epsilon) \leq \delta$. Expanding this we get

$$\sum_k \mathbb{P}\left(\exists \bar{x} \in B_{\ell_0}(x, \epsilon) \text{ such that } f(\bar{x}) \neq k\right) \mathbb{P}(y = k) \leq \delta.$$

In other words, there exists a class $k'$ satisfying $q_{k'}\left(\{x \in \mathcal{X} : \exists \bar{x} \in B_{\ell_0}(x, \epsilon) \text{ such that } f(\bar{x}) \neq k'\}\right) \leq \delta$. Defining the unsafe set for the class $k'$ as $U_{k'} = \{x \in \mathcal{X} : \exists \bar{x} \in B_{\ell_0}(x, \epsilon) \text{ such that } f(\bar{x}) \neq k'\}$, we have shown

$$q_{k'}(U_{k'}) \leq \delta. \tag{3}$$

Define $A_{k'} \subset \mathcal{X}$ to be the region where $f$ predicts $k'$, *i.e.*, $A_{k'} = \{x \in \mathcal{X} : f(x) = k'\}$. Further, for any set $Z$ define $Z^{+\epsilon}$ to be all the points in the domain $\mathcal{X}$ which are at most $\epsilon$ away from $Z$ in $\ell_0$ distance, *i.e.*, $Z^{+\epsilon} = \{x \in \mathcal{X} : \exists \bar{x} \in Z \text{ such that } \|x - \bar{x}\|_0 \leq \epsilon\}$ Then, we have

$$
\begin{aligned}
U_{k'} &= \{x \in \mathcal{X} : \exists \bar{x} \text{ such that } \|x - \bar{x}\|_0 \leq \epsilon, f(\bar{x}) \neq k'\} \\
&= \{x \in \mathcal{X} : \exists \bar{x} \in (\mathcal{X} \setminus A_{k'}) \text{ such that } \|x - \bar{x}\|_0 \leq \epsilon\} \\
&= (\mathcal{X} \setminus A_{k'})^{+\epsilon}.
\end{aligned}
$$

Now, we will use measure concentration on the unit cube from Talagrand (1995, Proposition 2.1.1):

**Lemma 2.3** (Proposition 2.1.1 in Talagrand (1995)). *For $B \subseteq [0,1]^n$, $\text{dist}(x, B) = \min_{z \in B} \|x - z\|_0$, any measure $\mu$ on $[0,1]$, we have*

$$
\mathbb{P}_{x \sim \mu^n}(\text{dist}(B, x) \geq t) \leq \frac{1}{\mathbb{P}_{x \sim \mu^n}(x \in B)} \exp(-t^2/n).
$$

Note that since the domain $[0,1]^n$ has $n$-dimensional volume 1, *i.e.*, $\text{Vol}([0,1]^n) = 1$, the uniform measure of any set $\mu^n(B) = \text{Vol}(B)$, for $B \subseteq [0,1]^n$. Substituting $B = \mathcal{X} \setminus A_{k'}$, $t = \epsilon$, $\mu = \text{Unif}([0,1])$, in Lemma 2.3, we obtain

$$
\text{Vol}(\mathcal{X} \setminus A_{k'})^{+\epsilon} \geq 1 - \frac{\exp(-\epsilon^2/n)}{\text{Vol}(\mathcal{X} \setminus A_{k'})}.
$$

Using $\text{Vol}(\mathcal{X} \setminus U_{k'}) = 1 - \text{Vol}(\mathcal{X} \setminus A_{k'})^{+\epsilon}$, we obtain

$$
\text{Vol}(\mathcal{X} \setminus U_{k'}) \leq \frac{\exp(-\epsilon^2/n)}{\text{Vol}(\mathcal{X} \setminus A_{k'})}. \tag{4}
$$

Finally, combining (3), (4), and taking $S = \mathcal{X} \setminus U_{k'}$, we have

$$
q_{k'}(S) \geq 1 - \delta, \qquad \text{Vol}(S) \leq C \exp(-\epsilon^2/n),
$$

where $C = \frac{1}{1 - \text{Vol}(A_{k'})}$, showing that $q_{k'}$ is $(C, \epsilon^2/n, \delta)$-localized. If the classes were balanced, repeating the above argument for each class shows that $q_k$ is $(C, \epsilon^2/n, K\delta)$-localized for all $k \in \mathcal{Y}$ for $C_{\max} = \max_{k'}(1/(1 - \text{Vol}(A_{k'})))$. $\qquad \square$

**Discussion on Theorem 2.2.** A few comments are in order for the above result.

1. Theorem 2.2 demonstrates that whenever a non-constant $\ell_0$ robust classifier exists for a data distribution, this distribution must be localized. This could be instantiated for real data sets like ImageNet to obtain interesting observations about the underlying distribution. For instance, humans are robust to perturbation of a few pixels to any image in ImageNet. Then, Theorem 2.2 tells us that ImageNet is localized. Note, however, that the localization *parameters* (*i.e.*, $C, \epsilon, \delta$ for the human classifier) are unknown.

2. The localization parameters in Theorem 2.2 are different than the concentration parameters in Pal et al. (2023, Theorem 2.1). Specifically, Pal et al. (2023, Theorem 2.1) shows that $(C, n\epsilon, \delta)$-concentration is a necessary condition for $\ell_2$-robustness under Definition 2.1, and we will now show that $(C, \epsilon^2/n, \delta)$-localization is a necessary condition for $\ell_0$-robustness. This demonstrates that the existence of a classifier robust to $\ell_0$ classifier implies a different kind of localization of the data distribution than robustness to $\ell_2$ perturbations. While Pal et al. (2023) assume that their data lies in a unit $\ell_2$ ball with adversarial perturbation strength $\epsilon \in [0,1]$, we assume that our data lies in a unit $\ell_\infty$ ball and with perturbation strength $\epsilon \in \{0, 1, 2, \ldots, n\}$. As such a direct comparison of the parameters is not immediate as our work deals with objects very different from Pal et al. (2023).

3. Theorem 2.2 suggests that for obtaining $\ell_0$ robust classifiers, we should try to find and classify over the sets that the distribution localizes on. This is a significant departure from the existing literature on $\ell_0$-robust classifiers Levine & Feizi (2020a); Jia et al. (2022); Hammoudeh & Lowd (2023), and indeed, we will obtain a classifier in Section 4 that respects such geometry.

We have now demonstrated that localization is a necessary condition for the existence of a classifier robust to perturbations bounded in the $\ell_0$ distance, *i.e.*, perturbations having a small support. Inspired by the investigations in Pal et al. (2023), we will now consider whether this condition is also sufficient.

## 3 $d$-Strong Localization implies Existence of a $d$-Robust Classifier

Localization of the data distribution ensures that each class conditional concentrates on a small volume subset of $\mathcal{X}$. However, as noted in Pal et al. (2023), these subsets might intersect too much, in which case there might not exist a classifier with low standard risk, *i.e.*, $R_{\ell_0}(f, 0)$. Hence, one cannot expect localization to be sufficient for the existence of a classifier with low robust risk, *i.e.*, $R_{\ell_0}(f, \epsilon)$ with $\epsilon > 0$. However, if these subsets were *separated* enough, then one can expect to use them to build a robust classifier. Indeed, we will now formalize this intuition to obtain a condition stronger than localization, which will be shown to be sufficient for the existence of a robust classifier.

**Definition 3.1** ($d$-Strongly Localized Distributions, generalizing Pal et al. (2023))**.** A distribution $p$ is said to be $(\epsilon, \delta, \gamma)$-strongly-localized with respect to a distance $d$, if each class conditional distribution $q_k$ localizes over the set $S_k \subseteq \mathcal{X}$ such that $q_k(S_k) \geq 1 - \delta$, and $q_k \left( \bigcup_{k' \neq k} S_{k'}^{+2\epsilon} \right) \leq \gamma$, where $S^{+\epsilon}$ denotes the $\epsilon$-expansion of the set $S$ in $d$, *i.e.*, $S^{+\epsilon} = \{x \colon \exists \bar{x} \in S \text{ such that } d(x, \bar{x}) \leq \epsilon\}$.

With the above definition, we will now obtain a generalization of Pal et al. (2023, Theorem 3.1) to an arbitrary distance $d$:

**Theorem 3.2.** *If $p$ is $(\epsilon, \delta, \gamma)$-strongly localized with respect to a distance $d$, then there exists a classifier $f$ such that $R_d(f, \epsilon) \leq \delta + \gamma$.*

*Proof.* At a high level, we will construct a classifier $g$ that predicts the label $k$ over an $\epsilon$-expansion of the set $S_k$ on which the class conditional $q_k$ localizes. We will then "shave off" some regions from each $S_k$ to ensure $g$ is well defined. For the rest of the input space $\mathcal{X}$ we will predict an arbitrary label, as we incur at most $\gamma$ in robust risk. Our construction of the robust classifier $f$ is same as that in Pal et al. (2023), extended to general $d$. However, bounding the robust risk of $f$ needs technical innovations, since we are bounding the robust risk with respect to a general distance $d$, as opposed to the $\ell_2$ norm in Pal et al. (2023).

For each $k \in \{1, 2, \ldots, K\}$, let $S_k$ be the set over which the conditional density $q_k$ is localized, i.e., $q_k(S_k) \leq 1 - \delta$. Define $S^{+\epsilon}$ to be the $\epsilon$-expansion of the set $S$, as $S^{+\epsilon} = \{x \colon \exists x' \in S, d(x, x') \leq \epsilon\}$. Define $C_k$ to be the $\epsilon$-expanded version of the localized region $S_k$ but removing the $\epsilon$-expanded version of all other regions $S_{k'}$, as

$$C_k = \left( S_k^{+\epsilon} \setminus \cup_{k' \neq k} S_{k'}^{+\epsilon} \right) \cap \mathcal{X}.$$

Similar to the construction in Pal et al. (2023), we will use these regions to define the classifier $f \colon \mathcal{X} \to \{1, 2, \ldots, K\}$ as

$$f(x) = \begin{cases} 1, & \text{if } x \in C_1 \\ 2, & \text{if } x \in C_2 \\ \vdots \\ K, & \text{if } x \in C_K \\ 1, & \text{otherwise} \end{cases}.$$

We will now show that $R_d(f, \epsilon) \leq \delta + \gamma$, which can be recalled to be

$$R_d(f, \epsilon) = \sum_k q_k(U_k) p_Y(y = k), \tag{5}$$

where the $q_k$ mass in (5) is over the set of all points $x \in \mathcal{X}$ that admit an $\epsilon$-adversarial example for the class $k$, defined as

$$U_k = \{x \in \mathcal{X} \colon \exists \bar{x} \in B_d(x, \epsilon) \cap \mathcal{X} \text{ such that } f(\bar{x}) \neq k\}. \tag{6}$$

As we saw earlier in the proof of Theorem 2.2, $U_k = (\mathcal{X} \setminus C_k)^{+\epsilon} \cap \mathcal{X}$. We will obtain an upper bound on $q_k(U_k)$, which will in turn give us an upper bound on $R_d(f, \epsilon)$.

Let $A = S_k^{+\epsilon} \cap \mathcal{X}$ and $B = \cup_{k' \neq k} S_{k'}^{+\epsilon}$. As $C_k = A \setminus B$, we have

$$
\begin{aligned}
\mathcal{X} \setminus C_k &= \mathcal{X} \cap (A \cap B^c)^c \\
&= \mathcal{X} \cap (A^c \cup B) \\
&= (\mathcal{X} \cap A^c) \cup (\mathcal{X} \cap B) \\
&= \left( \mathcal{X} \cap \left( S_k^{+\epsilon} \right)^c \right) \cup \left( \cup_{k' \neq k} (\mathcal{X} \cap S_{k'}^{+\epsilon}) \right).
\end{aligned}
$$

Then, we can expand $(\mathcal{X} \setminus C_k)^{+\epsilon}$

$$
\left( \mathcal{X} \cap \left( S_k^{+\epsilon} \right)^c \right)^{+\epsilon} \cup \left( \cup_{k' \neq k} (\mathcal{X} \cap S_{k'}^{+\epsilon})^{+\epsilon} \right),
$$

from the property $(U \cup V)^{+\epsilon} = U^{+\epsilon} \cup V^{+\epsilon}$. Now, since all the mass of $q_k$ lies in $\mathcal{X}$, i.e., $q_k(\mathcal{X}) = 1$, we have $q_k(\mathcal{X} \cap V) = q_k(V)$ for any set $V$. Applying this, we have

$$
\begin{aligned}
q_k(U_k) &= q_k(\mathcal{X} \setminus C_k)^{+\epsilon} \\
&\leq q_k \left( \mathcal{X} \cap \left( S_k^{+\epsilon} \right)^c \right)^{+\epsilon} + q_k \left( \cup_{k' \neq k} (\mathcal{X} \cap S_{k'}^{+\epsilon})^{+\epsilon} \right) \\
&\leq q_k \left( \left( S_k^{+\epsilon} \right)^c \right)^{+\epsilon} + q_k \left( \cup_{k' \neq k} (S_{k'}^{+\epsilon})^{+\epsilon} \right).
\end{aligned}
$$

Now applying Lemma A.1 we have $\left( \left( S_k^{+\epsilon} \right)^c \right)^{+\epsilon} = \left( \left( S_k^{+\epsilon} \right)^{-\epsilon} \right)^c$. Again from Lemma A.1 we know that $(V^{+\epsilon})^{-\epsilon} \supseteq V$ for any set $V$. Hence, we have $\left( \left( S_k^{+\epsilon} \right)^{-\epsilon} \right)^c \subseteq S_k^c$. Continuing,

$$
\begin{aligned}
q_k(U_k) &\leq q_k(S_k^c) + q_k \left( \cup_{k' \neq k} S_{k'}^{+2\epsilon} \right) \\
&\leq \delta + \gamma,
\end{aligned}
$$

Finally, as $\sum_k p_Y(y = k) = 1$, from (6) we have $R_d(f, \epsilon) \leq \delta + \gamma$. $\qquad \square$

We note that (Pal et al., 2023, Theorem 3.2) follows as a direct corollary of our result Theorem 3.2 by taking $d$ to be the $\ell_2$ distance.

**Implications for Existing Impossibility Results.** In our setting, Shafahi et al. (2018) prove that for any classifier $f \colon \mathcal{X} \to \{1, 2, \ldots, K\}$ for any class $k$ with $P(Y = k) \leq 1/2$, any point $x \sim q_k$ is either mis-classified, or admits an $\epsilon$-adversarial example with probability at least

$$
1 - \beta_{q_k} \exp \left( -\epsilon^2 / n \right), \tag{7}
$$

where $\beta_{q_k} = 2 \sup_x q_k(x)$ depends on the class conditional $q_k$. When $q_k$ is localized, $\beta_{q_k}$ can grow faster than $\exp \left( -\epsilon^2 / n \right)$, making the lower bound vacuous. This implies that *for localized data-distributions there is no impossibility, and there is a wide class of high-dimensional classification problems for which robust classifiers exist.* We now provide a concrete example.

**Example 3.1.** *Let us consider a problem with 2 classes defined by the distribution $p$ such that $P(Y = 0) = P(Y = 1) = 1/2$, the class conditional $q_1 = P(X|Y = 1) = \mathrm{Unif}(B_{\ell_\infty}(\mathbf{1}, \epsilon))$, and similarly $q_2 = P(X|Y = 2) = \mathrm{Unif}(B_{\ell_\infty}(-\mathbf{1}, \epsilon))$. For this distribution, $\beta_{q_1} = \beta_{q_2} = \exp(n)$, and the lower bound (7) becomes vacuous for $\epsilon \leq \sqrt{n}$ as*

$$
1 - \beta_{q_k} \exp \left( -\epsilon^2 / n \right) = 1 - 2 \exp(-\epsilon^2/n + n) \leq 0.
$$

Even though Example 3.1 is quite simple, the construction of small $\ell_\infty$ balls in the input space containing most of the mass of the distribution is quite general, and depicts a wide class of data-distributions where existing impossibility results are vacuous. We will now demonstrate that these general theoretical ideas lead to practical $\ell_0$ robust classifiers.

## 4 $\ell_0$-**Adversarially Robust Classification via the Box-NN classifier**

In this section, our aim will be to derive a $\ell_0$-robust classifier by utilizing the geometry exposed by Theorem 3.2. To this end, we will first investigate how a robust classifier looks like for a simple 2-class problem in 3-dimensions. This will motivate a general form of a classifier whose decision regions are axis-aligned cuboids, or boxes. Finally, we will generalize this classifier to obtain a $\ell_0$-robust classifier and derive corresponding $\ell_0$ certificates.

### 4.1 Development and Robustness Certification

Consider $n = 3$, and say there are two classes, CAT and DOG, defining conditional distributions $q_1$ and $q_2$, strongly localized over $S_1$ and $S_2$ respectively, such that $q_1(S_2^{+1}) = 0$ and $q_2(S_1^{+1}) = 0$. In such a situation, Theorem 3.2 (invoked with $\epsilon = 1$) constructs a robust classifier $f_A$ as the following:

$$f_A(x) = \begin{cases} \text{dog}, & \text{if } x \in S_1^{+1} \\ \text{cat}, & \text{if } x \in S_2^{+1} \\ \text{cat}, & \text{otherwise.} \end{cases}$$

However, in practice, $S_1, S_2$ might be very complex, and hence $f_A$ might be computationally hard to evaluate. For instance, Fig. 1 shows an illustration where these sets (shaded green and orange) have complicated shapes.

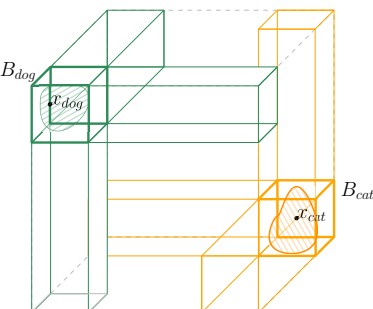

Figure 1: $S_1$ is the green shaded region around $x_{dog}$, where the class dog is localized, and $S_2$ is the orange shaded region around $x_{cat}$, where the class cat is localized.

From Fig. 1, we see that the classifier $f_A$ is robust to 1-pixel perturbations whenever $x \in S_1$ or $x \in S_2$, as Theorem 3.2 predicts. More importantly, we see that a perturbation of a single pixel of any $x_{cat} \in S_2$ lies within the union of the orange cuboids. In other words, $\{x' \in [0,1]^3 : \|x - x\|_0 \leq 1, x \in S_1\} = S_1^{+1} \subseteq$ ORANGE, and similarly for the dog class. Furthermore, we see that the intersection of these orange cuboids is given by the cube $B_{cat}$. We can see that for any $x \in B_{cat}$, no single-pixel perturbation $v$ can take $x + v$ outside the orange region ORANGE, and similarly for the dog class. However, $B_{cat}, B_{dog}$ are very efficiently described, they are simply axis-aligned polyhedra enclosing $S_2$ and $S_1$ respectively. This motivates our modified classifier $f_B$,

$$f_B(x) = \begin{cases} \text{dog}, & \text{if } x \in B_{dog}^{+1} \\ \text{cat}, & \text{if } x \in B_{cat}^{+1} \\ \text{cat}, & \text{otherwise.} \end{cases}$$

While $f_B$ is efficient to describe, it ignores a large portion of the input region outside the green and the orange cuboids, *i.e.*, $\mathcal{X} \setminus B_{dog}^{+1} \cup B_{cat}^{+1}$, by making the constant prediction cat in this region. We can further extend $f_B$ to attempt to correctly classify those regions as well, by computing $\ell_0$ distances to our boxes $B_{cat}, B_{dog}$, as

$$f_C(x) = \underset{y \in \{\text{cat}, \text{dog}\}}{\arg\min} \text{dist}(x, B_y),$$

where

$$\text{dist}(x, S) = \min_v \|v\|_0 \text{ sub. to } x + v \in S \tag{8}$$

gives the minimum number of pixel changes needed to get from $x$ to $S$. While solving (8) is computationally hard for general $S$, the following lemma shows that for our axis-aligned boxes $B$, (8) can be computed efficiently, in closed form. The proofs of all our results can be found in Appendix A.

**Lemma 4.1** ($\ell_0$ distance to axis-aligned boxes)**.** *For an axis aligned box $B(a, b)$ specified as $B(a, b) = \{x : a \leq x \leq b\}$, where $a, b, x \in \mathbb{R}^n$, and all inequalities are element-wise, we have*

$$\text{dist}(x, B(a, b)) = \sum_{i=1}^{n} \mathbf{1}\left(x_i \notin [a_i, b_i]\right),$$

*which can be computed in $O(n)$ operations.*

For real data distributions, however, having a single box per class would be overly simplistic and not provide good accuracy. Thus, we generalize $f_C$ to our Box-NN classifier operating on boxes $\mathcal{B} = \{B_1, B_2, \ldots, B_M\}$, such that we have an label $y_m \in \{1, 2, \ldots, K\}$ associated with each $B_m$. Our Box-NN classifier is then defined as

$$\text{Box-NN}(x, \mathcal{B}) = y_{m^\star}, \text{ where } m^\star = \arg\min_m \text{dist}(x, B_m).$$

Note that, so far, we have not described how these boxes $\mathcal{B}$ are learned from data. This will be the subject of Section 4.2 and onward. We can now obtain a $\ell_0$ robustness certificate for Box-NN via the following Theorem.

**Theorem 4.2** (Robustness Certificate for Box-NN)**.** *Given a set of boxes $\mathcal{B}$ and their associated labels $\{y_m\}_{m=1}^{M}$, define*

$$m^\star = \arg\min_m \text{dist}(x, B_m), \quad d_1 = \text{dist}(x, B_{m^\star}),$$

*and*

$$d_2 = \min_{m:\, y_m \neq y_{m^*}} \text{dist}(x, B_m).$$

*Then, with $\text{margin}(x) \overset{\text{def}}{=} d_2 - d_1$, we have $\text{Box-NN}(x, \mathcal{B}) = \text{Box-NN}(x', \mathcal{B})$ whenever $\|x' - x\|_0 < \text{margin}(x)/2$.*

**Key Intuition.** Our robust classifier Box-NN is essentially a generalization of the nearest-neighbor classifier to a nearest-box classifier, specifically suited to $\ell_0$ metrics. This simple form turns out to be the right choice, in the sense of the theoretical motivation of our previous section, for defending against sparse perturbations. As we will shortly see, Box-NN also empirically produces better certificates than prior work in several regimes.

Having developed the geometric intuition and the theoretical robustness guarantees for Box-NN, we will now describe how we learn our classifier from data, and the associated challenges.

## 4.2 Learning Box-NN from Data

In this section, we are concerned with learning boxes $\{B_m\}$ and their associated labels $\{y_m\}$, such that Box-NN obtains a high accuracy under sparse adversarial perturbations. For the rest of this section, we will refer to the classifier Box-NN as $f_\theta$, with the learnable parameters $\theta = \{a_k, b_k, y_k\}_{k=1}^{M}$ following the notation in Lemma 4.1.

The quantity we are interested in maximizing is the robust accuracy, defined as $1 - R_{\ell_0}(f_\theta, \epsilon)$ following our notation in Section 2. As we do not have access to the data distribution, we will instead be concerned with maximizing the empirical robust accuracy $\text{RobustAcc}(f_\theta, \epsilon)$ defined over a set of samples $\{x_i, y_i\}_{i=1}^{N}$ given by

$$\frac{1}{N} \sum_{i=1}^{N} \mathbf{1}\left[\forall x' : \|x' - x_i\|_0 \leq \epsilon, \ f_\theta(x') = y_i\right]. \tag{9}$$

The objective in (9) is a complicated object, and direct maximization w.r.t. $\theta$ is challenging. In the following, we will first lower bound (9) and then use several optimization tricks to efficiently maximize this lower bound.

Recall from Theorem 4.2 that $f_\theta(x) = f_\theta(x')$ for all $x'$ satisfying $\|x - x'\|_0 \leq C_\theta(x) \overset{\text{def}}{=} \text{margin}(x)/2$, where $C_\theta$ is a pointwise certificate (at $x$) of robustness for $f_\theta$. With this, we have the certified accuracy lower bound $\text{RobustAcc}(f_\theta, \epsilon) \geq \text{CertAcc}(f_\theta, \epsilon)$ defined as

$$\text{CertAcc}(f_\theta, \epsilon) \overset{\text{def}}{=} \frac{1}{N} \sum_{i=1}^{N} \mathbf{1}[f_\theta(x_i) = y_i] \cdot \mathbf{1}[C_\theta(x_i) \geq \epsilon]. \tag{10}$$

We will take a gradient based optimization approach to maximize (10) over $\theta$. However, since the gradients of $\mathbf{1}[\cdot]$ are zero almost everywhere (and discontinuous otherwise), we will progressively relax the indicators in (10). To this end, we maximize the integral of $\text{CertAcc}(f_\theta, \epsilon)$ over all $\epsilon \geq 0$ instead of treating it point-wise[4], leading to the objective

$$L_1(\theta) = \frac{1}{N} \sum_{i=1}^{N} \mathbf{1}[f_\theta(x_i) = y_i] \cdot C_\theta(x_i). \tag{11}$$

**Approximating** min. Recall from Theorem 4.2 that the margin involves the min function,

$$\text{margin}(x) = \min_m \text{dist}(x, B_m) - \min_{m:\, y_m \neq y_{m^\star}} \text{dist}(x, B_m).$$

The gradient of min w.r.t. its input $(c_1, \ldots, c_M)$ is extremely sparse[5], and hence a very small number of parameters $\theta_i$ are updated at each step of gradient descent using gradients of (11). As a result, optimization is extremely slow. We remedy this by using a soft approximation to min which has dense gradients,

$$\min{}_\tau \{c_1, \ldots, c_M\} \overset{\text{def}}{=} \sum_{m=1}^{M} c_m \frac{\exp(-\tau c_m)}{\sum_j \exp(-\tau c_j)}, \tag{12}$$

where $\tau$ is a parameter that approximately controls the sparsity of the gradients. The function $\min{}_\tau$ is equal to min in the limit $\tau \to \infty$, and reduces to the average when $\tau = 0$.

Furthermore, we find that for many data points $x_i$, a small number of boxes $m$ contribute a lot to the final loss due to large distances $\text{dist}(x_i, B_m)$. As a result, learning is slow for parameters corresponding to the remaining boxes. To prevent such imbalance, we clip the certificates to 50. With these approximations, we obtain

$$L_2(\theta) = \frac{1}{N} \sum_{i=1}^{N} \mathbf{1}[f_\theta(x_i) = y_i] \cdot \tilde{C}_\theta(x_i),$$

where $\tilde{C}_\theta(x)$ is defined as

$$\min \left( \min{}_\tau_m \text{dist}(x, B_m) - \min{}_\tau_{m:\, y_m \neq y_{m^\star}} \text{dist}(x, B_m), 50 \right). \tag{13}$$

**Relaxing Indicator Functions.** Now observe that $L_2$ is still a function of indicator functions, due to the dist function in (13), which was derived in Lemma 4.1 to be $\text{dist}(x, B(a, b)) = \sum_{i=1}^{n} \mathbf{1}(x_i \notin [a_i, b_i])$. Again, as the gradients of $\mathbf{1}[\cdot]$ are zero almost everywhere, we perform a conical approximation to $\mathbf{1}(x_i \notin [a_i, b_i])$ which has non-zero gradients:

$$\text{conical}(x, a_i, b_i) \overset{\text{def}}{=} \max(a_i - x, 0) + \max(x - b_i, 0). \tag{14}$$

Finally, we replace the indicator $\mathbf{1}[f_\theta(x_i) = y_i]$ in $L_2$ by $s_i$, where $s_i = +1$ if $f(x_i) = y_i$, and $s_i = -1$ otherwise, to have the misclassified data-points contribute to the loss. These modifications lead to our final objective $L(\theta)$.

---

[4] *i.e.*, $\int_{\epsilon \geq 0} \mathbf{1}[\epsilon \leq \alpha] d\epsilon = \alpha$

[5] $\nabla_c \min(c_1, c_2, \ldots, c_m) = (0, \ldots, 0, 1, 0, \ldots, 0) = e_{j^\star}$, where $e_j$ is the $j^{\text{th}}$ standard basis vector, and $j^\star = \arg\min_j c_j$.

**Improving Initialization.** We initialize $\theta$ by using a set of boxes defined from the data. This is done by first drawing a subset $T$ of size $M$ uniformly at random from the training data-points, and then initializing $\theta$ with axis-aligned boxes centered at these data-points, as $\{(B(x - 0.1, x + 0.1), y) : (x, y) \in T\}$, where the scalar is added to, or subtracted from, every co-ordinate of the vector. Having described all the tricks used for optimizing Box-NN, we now proceed to performing an empirical evaluation in Section 5. For the interested reader, we also ablate over the training strategies mentioned above in Appendix B.2.

## 5 Empirical Evaluation

In this section, we will briefly describe existing methods for probabilistic $\ell_0$ certification, (Levine & Feizi, 2020b) and (Jia et al., 2022) as well as deterministic $\ell_0$ certification (Hammoudeh & Lowd, 2023), and then empirically compare our (deterministic) $\ell_0$ certified defense Box-NN to these approaches.

Levine & Feizi (2020b) and Jia et al. (2022) extend the technique of randomized smoothing (Cohen et al., 2019) to randomized ablation (RA), where given any classifier $f$ (*e.g.*, a neural network), they produce a smoothed classifier by zeroing out $\rho$ pixels uniformly at random:

$$\text{RA}_\rho(x) = \arg\max_k \mathop{\mathbb{P}}_{v \sim \text{Unif}(V_\rho)} (f(x \odot v) = k), \tag{15}$$

where $V_\rho = \{v \in \{0, 1\}^n : \|v\|_0 = \rho\}$ is the discrete set of all binary vectors of length $n$ having exactly $\rho$ ones, and $\odot$ denotes the Hadamard product. For this construction in (15), a counting argument leads to the robustness certificate in Levine & Feizi (2020b), which we compare to in Fig. 2. A more complicated analysis based on the Neyman-Pearson lemma leads to a tighter certificate in Jia et al. (2022), which is also included in our comparison in Fig. 3 (left), denoted by $\text{RA}_\rho^\text{B}$. Both these certificates are randomized, *i.e.*, they hold with a confidence $1 - \alpha$, where $\alpha, \rho$ are hyper-parameters that trade-off benign accuracy to robustness, and can be chosen empirically. According to standard practice, we fix $\alpha = 0.05$ and produce plots for varying $\rho$. The interested reader can refer to Appendix B.3 for a description of the certification procedures developed in Levine & Feizi (2020b); Jia et al. (2022).

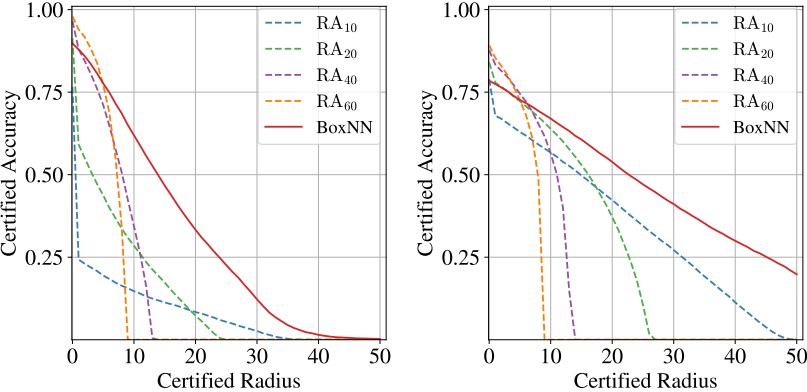

Figure 2: Comparison of Randomized Ablation (Levine & Feizi, 2020b) to our method Box-NN on the MNIST (left) and FashionMNIST (right) datasets. In each figure, the dotted lines correspond to different hyperparameter settings $\rho$. Details in text.

More recently, given any classifier $f$, Hammoudeh & Lowd (2023) produce a deterministic $\ell_0$ certified classifier $g$ by partitioning the set of pixels $\{1, 2, \ldots, n\}$ into disjoint partitions $\mathcal{V}$, and then producing the majority prediction of $f$ over $\mathcal{V}$:

$$\text{FPA}(x) = \text{Majority}\{f(x_V)\}_{V \in \mathcal{V}}, \tag{16}$$

where $f(x_V)$ is defined as the prediction of $f$ obtained after zeroing out the pixels in $x$ not in $V$. Hammoudeh & Lowd (2023) then produce a certificate by counting the difference in the votes of the majority label to the

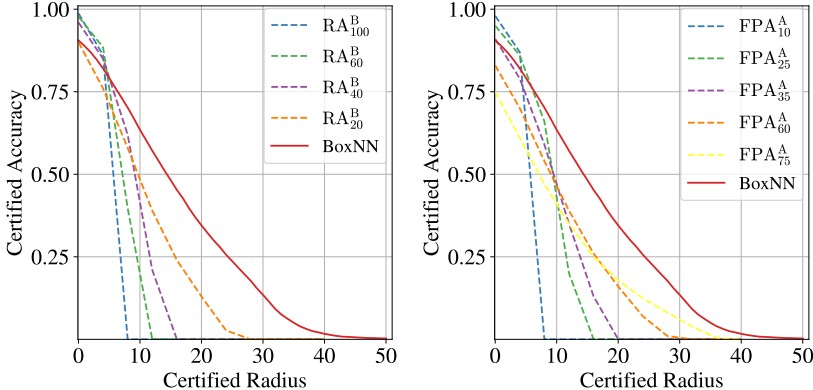

Figure 3: Comparison of Jia et al. (2022) (left) and Hammoudeh & Lowd (2023) (right) to our method Box-NN on the MNIST dataset. The dotted lines correspond to different settings for the hyperparameter $\rho$. Details are mentioned in text.

Table 1: Comparison of the median certified radius $\bar{r}$ obtained by our Box-NN to the best hyperparameter settings for prior work.

| Dataset | Method | $\bar{r}$ |
|---|---|---|
| | Box-NN | **13** |
| | RA Levine & Feizi (2020b) | 8 |
| MNIST | RA$^B$ Jia et al. (2022) | 10 |
| | FPA$^A$ Hammoudeh & Lowd (2023) | 9 |
| | FPA$^B$ Hammoudeh & Lowd (2023) | 12 |
| FMNIST | Box-NN | **22** |
| | RA Levine & Feizi (2020b) | 16 |

runner-up label in (16). In Fig. 3 (right), we compare to the best performing strategy for constructing $\mathcal{V}$ in (Hammoudeh & Lowd, 2023) named "strided" and denoted by FPA$^A_\rho$, where equally spaced pixels are selected for each partition, *i.e.*, $\mathcal{V} = \{j : j \equiv t - 1 \bmod \rho\}_{t=0}^{\rho-1}$. Here $\rho$ is a hyper-parameter as earlier, and we vary $\rho$ to produce the plots in Fig. 3[6]. Note that (Hammoudeh & Lowd, 2023) also obtain an improved certificate by using an aggregation more complicated than the majority vote (called "FPA with run-off elections"), which is compared to in Fig. 10, where it is denoted by FPA$^B_\rho$. The interested reader can refer to Appendix B.3 for more details.

**Results** Recall from Section 4.2 Eq. (10) that the certified accuracy of a classifier $g$ against $\epsilon$-bounded adversarial perturbations, CertAcc$(g, \epsilon)$, can be obtained given a point-wise certificate $C$ for $g$. For each of the methods described so far, we plot CertAcc against $\epsilon$ using the corresponding robust classifier $g$ and the certificate $C$ over samples from the test set of the datasets mentioned.

A commonly used metric for comparing certified accuracy curves adopted in the literature (Levine & Feizi, 2020b; Jia et al., 2022; Hammoudeh & Lowd, 2023) is the median certified radius, which is the largest perturbation strength under which a classifier is certified to have atleast 50% robust accuracy. As can be seen in Table 1, our method Box-NN *outperforms all existing methods under all hyperparameter settings on this metric.*

The median certified radius captures a small slice of the full certified accuracy curve, which provides a complete picture. Observe that the dotted curves in Figs. 2 and 3 remain lower than our red curve except at small attack strengths. This shows that Box-NN is able to produce better certificates at most radii, and trades-off

---

[6]We use the results reported in Hammoudeh & Lowd (2023, Table 27) given that no public implementation of the method is available, to the best of our knowledge.

robustness at higher radii for benign accuracy at small radii. Without any dedicated hyper-parameter tuning, Box-NN dominates any single dotted curve for a large range of attack strengths, demonstrating that certified defenses closely utilizing properties of the data-distribution can outperform complicated ensembling-based defenses which ignore properties of the data.

## 6 Conclusion, Limitations and Future Work

In this work, we developed a theoretical to exploit properties of the data distribution for robustness against sparse adversarial attacks. We showed that data localization – the property that a data distribution $p$ places most of its mass on very small volume sets in the input space – characterizes the existence of a $\ell_0$-robust classifier for $p$. Following this theory, we developed a defense against sparse adversarial attacks, and derived a corresponding robustness certificate. We showed that this certificate empirically improves upon existing state-of-the-art in several broad regimes.

The primary limitation of our work is the difficulty in efficiently learning classifiers that have axis-aligned decision regions. While we are able to successfully employ several optimization tricks for datasets like MNIST and Fashion MNIST, the task becomes harder on more complicated datasets, *even though the geometry required for the underlying data-distribution remains the same due to our general theoretical results*. These optimization difficulties mostly stem from the strict requirement of axis-aligned boxes for our distance computation in Lemma 4.1. In the future, we hope to trade-off efficiency in the distance computation in favor of richer decision boundaries that can be learnt efficiently and generalize well.

### Acknowledgements

The authors thank the anonymous reviewers for their valuable suggestions, which helped improve this manuscript. This work was supported by Amazon (via the JHU-Amazon Initiative for Interactive AI, AI2AI), DARPA (via GARD HR00112020010 and HR00112020004) and NSF (via Grant 2031985 and 2212457).

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

## A    Auxilliary Lemmas and Proofs

**Lemma A.1** (Properties of expansion and contraction, extending Pal et al. (2023))**.** *For a distance $d$, set $A \subseteq [0,1]^n$, define $A^{+\epsilon} = \{x \in [0,1]^n \colon \mathrm{dist}_d(x, A) \leq \epsilon\}$, and $A^{-\epsilon} = \{x \in [0,1]^n \colon B_d(x, \epsilon) \subseteq A\}$. Then, for $N, O \subseteq [0,1]^n$, we have*

1. *$(N \cap O)^{-\epsilon} = N^{-\epsilon} \cap O^{-\epsilon}$*

2. *$(N^c)^{-\epsilon} = (N^{+\epsilon})^c$, where $c$ denotes complement in $[0,1]^n$*

3. *$(N \setminus O)^{-\epsilon} = N^{-\epsilon} \setminus O^{+\epsilon}$*

4. *$(N \cup O)^{+\epsilon} = N^{+\epsilon} \cup O^{+\epsilon}$*

5. *$(N^{+\epsilon_1})^{+\epsilon_2} \subseteq N^{+(\epsilon_1 + \epsilon_2)}$*

*Proof.* The first four assertions of this Lemma are standard results in mathematical morphology, dealing with the erosion and dilation of sets, and are reproduced here from Pal et al. (2023) for clarity.

1. Let $M = N \cap O$.

$$M^{-\epsilon} = \{x \colon x \in M, B_d(x, \epsilon) \subseteq M\}$$
$$= \{x \colon x \in N, x \in O, B_d(x, \epsilon) \subseteq N, B_d(x, \epsilon) \subseteq O\} = N^{-\epsilon} \cap O^{-\epsilon}.$$

2. Let $M = N^c$.

$$M^{-\epsilon} = \{x \colon x \in M, B_d(x, \epsilon) \subseteq M\} = \{x \colon x \notin N, B_d(x, \epsilon) \subseteq N^c\}$$
$$= \{x \colon x \notin N, \forall x' \in B_d(x, \epsilon) \ x' \notin N\}$$
$$= \{x \colon \forall x' \in B_d(x, \epsilon) \ x' \notin N\}$$
$$\implies (M^{-\epsilon})^c = \{x \colon \exists x' \in B_d(x, \epsilon) \ x' \in N\}$$
$$= N^{+\epsilon}.$$

3. Let $M = N \setminus O$, we have $M^{-\epsilon} = (N \cap O^c)^{-\epsilon} = N^{-\epsilon} \cap (O^c)^{-\epsilon}$ by Property 1, and then $N^{-\epsilon} \cap (O^c)^{-\epsilon} = N^{-\epsilon} \cap (O^{+\epsilon})^c$ by Property 2.

4. Let $M = N \cup O$. We have $M^c = N^c \cap O^c$. Taking $\epsilon$-contractions, and applying the first and second properties, we get $M^{+\epsilon} = N^{+\epsilon} \cup O^{+\epsilon}$.

5. For a set $M$, and any $\epsilon_1 \geq 0, \epsilon_2 \geq 0$, we have

$$\left(M^{+\epsilon_1}\right)^{+\epsilon_2} \subseteq M^{+(\epsilon_1 + \epsilon_2)}.$$

The above property can be derived from the triangle inequality applied to $d$, as

$$\left(M^{+\epsilon_1}\right)^{+\epsilon_2} = \{x \colon \exists x' \in M^{+\epsilon_1}, \ d(x', x) \leq \epsilon_2\}$$
$$= \{x \colon \exists x' \in \mathcal{X}, x'' \in M, \ d(x', x) \leq \epsilon_2, d(x'', x') \leq \epsilon_1\}$$
$$\subseteq \{x \colon \exists x'' \in M, \ d(x'', x) \leq \epsilon_2 + \epsilon_1\} = M^{+(\epsilon_1 + \epsilon_2)}.$$

$\square$

**Lemma 4.1** ($\ell_0$ distance to axis-aligned boxes)**.** *For an axis aligned box $B(a, b)$ specified as $B(a, b) = \{x \colon a \leq x \leq b\}$, where $a, b, x \in \mathbb{R}^n$, and all inequalities are element-wise, we have*

$$\mathrm{dist}(x, B(a, b)) = \sum_{i=1}^{n} \mathbf{1}\left(x_i \notin [a_i, b_i]\right),$$

*which can be computed in $O(n)$ operations.*

*Proof.* For any given $x$, recall the definition of dist to be $\mathrm{dist}(x, B(a,b)) = \min_{y \in B(a,b)} \|x - y\|_0$. For any $y \in B(a,b)$ we have,

$$\|x - y\|_0 = \sum_{i=1}^{n} \mathbf{1}(x_i \neq y_i) \geq \sum_{i=1}^{n} \mathbf{1}(x_i \notin [a_i, b_i])\mathbf{1}(y_i \in [a_i, b_i]) = \sum_{i=1}^{n} \mathbf{1}(x_i \notin [a_i, b_i]) \tag{17}$$

The above implies $\min_{y \in B(a,b)} \|x - y\|_0 \geq \sum_{i=1}^{n} \mathbf{1}(x_i \notin [a_i, b_i])$. Then, consider $y^\star \in B(a,b)$ defined as

$$y_i^\star = \begin{cases} a_i & \text{if } x_i \notin [a_i, b_i] \\ x_i & \text{otherwise} \end{cases}. \tag{18}$$

We have $\|y^\star - x\|_0 = \sum_{i=1}^{n} \mathbf{1}(x_i \notin [a_i, b_i])$, which attains the lower bound on $\mathrm{dist}(x, B(a,b))$. The result follows. $\qquad\square$

**Theorem 4.2** (Robustness Certificate for BOX-NN)**.** *Given a set of boxes $\mathcal{B}$ and their associated labels $\{y_m\}_{m=1}^{M}$, define*

$$m^\star = \arg\min_m \mathrm{dist}(x, B_m), \quad d_1 = \mathrm{dist}(x, B_{m^\star}),$$

*and*

$$d_2 = \min_{m : y_m \neq y_{m^*}} \mathrm{dist}(x, B_m).$$

*Then, with* $\mathrm{margin}(x) \stackrel{\text{def}}{=} d_2 - d_1$, *we have* $\mathrm{BOX\text{-}NN}(x, \mathcal{B}) = \mathrm{BOX\text{-}NN}(x', \mathcal{B})$ *whenever* $\|x' - x\|_0 < \mathrm{margin}(x)/2$.

*Proof.* Let $x, x' \in \mathcal{X}$. Define $\mathcal{B}_1 = \{B_m : y_m = y_{m^\star}\}$, and $\mathcal{B}_2 = \{B_m : y_m \neq y_{m^\star}\}$. Further, define $\bar{d}_1, \bar{d}_2$ as

$$d_1(x') = \min_{B \in \mathcal{B}_1} \mathrm{dist}(x', B), \quad d_2(x') = \min_{B \in \mathcal{B}_2} \mathrm{dist}(x', B),$$

Our goal would be to demonstrate that as long as $\|x - x'\|_0 < \mathrm{margin}(x)/2$, we have $d_2(x') > d_1(x')$, implying that the prediction remains the same at $x'$. Consider any $B \in \mathcal{B}_2$, and apply the triangle inequality to get

$$\mathrm{dist}(x', B) + \|x - x'\|_0 \geq \mathrm{dist}(x, B), \tag{19}$$

where (19) can be seen as

$$\mathrm{dist}(x', B) + \|x - x'\|_0 = \min_{y \in B} \|y - x'\|_0 + \|x' - x\|_0 \geq \min_{y \in B} \|y - x\|_0 = \mathrm{dist}(x, B). \tag{20}$$

Further, taking a minimum on both sides of (19) over all $B \in \mathcal{B}_2$ leads to

$$d_2(x') + \|x - x'\|_0 \geq d_2 \tag{21}$$

Similarly, consider any $B \in \mathcal{B}_1$, and apply the triangle inequality to get

$$\mathrm{dist}(x, B) + \|x - x'\|_0 \geq \mathrm{dist}(x', B), . \tag{22}$$

Taking a minimum over both sides of (22) over all $B \in \mathcal{B}_1$ leads to

$$d_1 + \|x - x'\|_0 \geq d_1(x'). \tag{23}$$

Adding (21) and (23), we have

$$d_2(x') - d_1(x') + 2\|x - x'\|_0 \geq d_2 - d_1 \tag{24}$$

$$\implies d_2(x') - d_1(x') \geq \mathrm{margin}(x) - 2\|x - x'\|_0, \tag{25}$$

from where we can see that $d_2(x') - d_1(x') > 0$ whenever $\|x - x'\|_0 < \mathrm{margin}(x)/2$, as required. $\qquad\square$

## B  Additional Experiments

### B.1  Additional Details for Figures

We provide fine-grained numerical details for Figs. 2, 3 and 10. We sample the $x$-axis at equally spaced intervals of size 4 and report the certified accuracies at certified radii of $\{0, 4, 8, 12, 16, 20, 24, 28, 32, 36, 40\}$ in Tables 2 and 3.

Table 2: Certified Accuracy on MNIST corresponding to plots in Fig. 2 (left), Fig. 3, and Fig. 10.

| METHOD | CERTIFIED RADIUS | | | | | | | | | | |
|---|---|---|---|---|---|---|---|---|---|---|---|
| | 0 | 4 | 8 | 12 | 16 | 20 | 24 | 28 | 32 | 36 | 40 |
| BOX-NN | 89.74 | 80.68 | 68.61 | 55.63 | 43.95 | 33.34 | 24.71 | 16.54 | 8.26 | 3.57 | 1.48 |
| $\text{RA}_{10}$ | 76.62 | 20.81 | 16.36 | 13.1 | 10.8 | 8.39 | 5.99 | 3.72 | 1.47 | 0.09 | 0 |
| $\text{RA}_{20}$ | 91.33 | 47.19 | 33.69 | 23.49 | 15.15 | 7.3 | 0.49 | 0 | 0 | 0 | 0 |
| $\text{RA}_{40}$ | 96.51 | 76.08 | 50.76 | 16.03 | 0 | 0 | 0 | 0 | 0 | 0 | 0 |
| $\text{RA}_{60}$ | 97.96 | 83.84 | 34.45 | 0 | 0 | 0 | 0 | 0 | 0 | 0 | 0 |
| $\text{RA}_{10}^{\text{B}}$ | 98.75 | 86.1 | 0 | 0 | 0 | 0 | 0 | 0 | 0 | 0 | 0 |
| $\text{RA}_{20}^{\text{B}}$ | 97.78 | 88.45 | 39.75 | 0 | 0 | 0 | 0 | 0 | 0 | 0 | 0 |
| $\text{RA}_{40}^{\text{B}}$ | 96 | 85 | 62 | 21 | 0 | 0 | 0 | 0 | 0 | 0 | 0 |
| $\text{RA}_{60}^{\text{B}}$ | 90 | 76 | 58 | 39 | 24 | 12.98 | 2.7 | 0 | 0 | 0 | 0 |
| $\text{FPA}_{10}^{\text{A}}$ | 98 | 87 | 0 | 0 | 0 | 0 | 0 | 0 | 0 | 0 | 0 |
| $\text{FPA}_{25}^{\text{A}}$ | 95 | 86 | 66 | 20 | 0 | 0 | 0 | 0 | 0 | 0 | 0 |
| $\text{FPA}_{35}^{\text{A}}$ | 91 | 79 | 59 | 35 | 13 | 0 | 0 | 0 | 0 | 0 | 0 |
| $\text{FPA}_{60}^{\text{A}}$ | 83 | 70 | 54 | 39 | 26 | 16 | 7 | 1 | 0 | 0 | 0 |
| $\text{FPA}_{75}^{\text{A}}$ | 74.99 | 61 | 47 | 34.97 | 25 | 17.9 | 12.43 | 8.11 | 3.89 | 0.42 | 0 |
| $\text{FPA}_{10}^{\text{B}}$ | 99 | 87 | 0 | 0 | 0 | 0 | 0 | 0 | 0 | 0 | 0 |
| $\text{FPA}_{25}^{\text{B}}$ | 96 | 88 | 71 | 20 | 0 | 0 | 0 | 0 | 0 | 0 | 0 |
| $\text{FPA}_{35}^{\text{B}}$ | 93 | 83 | 67 | 44 | 14 | 0 | 0 | 0 | 0 | 0 | 0 |
| $\text{FPA}_{65}^{\text{B}}$ | 87 | 76 | 63 | 50 | 37 | 23 | 12.14 | 2.97 | 0 | 0 | 0 |
| $\text{FPA}_{75}^{\text{B}}$ | 81 | 68 | 56.44 | 44.65 | 34.68 | 25 | 17.82 | 11.09 | 5.28 | 0.45 | 0 |

Table 3: Certified Accuracy on Fashion-MNIST corresponding to plots in Fig. 2 (right).

| METHOD | CERTIFIED RADIUS | | | | | | | | | | |
|---|---|---|---|---|---|---|---|---|---|---|---|
| | 0 | 4 | 8 | 12 | 16 | 20 | 24 | 28 | 32 | 36 | 40 |
| BOX-NN | 78.43 | 73.84 | 69.31 | 64.44 | 59.26 | 53.86 | 48.39 | 43.43 | 38.73 | 34.05 | 29.97 |
| $\text{RA}_{10}$ | 78.96 | 64.28 | 59.15 | 53.86 | 48.32 | 42.32 | 36.07 | 30 | 24.08 | 17.69 | 11.31 |
| $\text{RA}_{20}$ | 84.05 | 73.65 | 67.54 | 60.05 | 50.35 | 37.17 | 18.03 | 0 | 0 | 0 | 0 |
| $\text{RA}_{40}$ | 87.7 | 77.1 | 65.4 | 39.9 | 0 | 0 | 0 | 0 | 0 | 0 | 0 |
| $\text{RA}_{60}$ | 89.22 | 76.54 | 48.89 | 0 | 0 | 0 | 0 | 0 | 0 | 0 | 0 |

## B.2 Ablation Study

We provide an ablation study on the various strategies used to effectively train BoxNN developed in Section 4.2. For each of the following certified accuracy plots in our ablation study, we fix all the other parameters at the values specified in Table 4 (these values correspond to those reported in the main text, and are explained in the following description), and vary the parameter specified. The red curve is the same in all the plots, and corresponds to the BoxNN plots in the main text.

Table 4: Default Parameters (red curve) for Ablation Study of BoxNN.

| min Approximation | Indicator Approximation | $\alpha$ Initialization | Boxes $M$ | Clipping $\beta$ | Optimizer |
|---|---|---|---|---|---|
| $\min_1$ | conical | 0.1 | 2500 | 50 | Vanilla SGD (0.2) |

1. **Indicator Relaxation.** Recall that we relax the indicator function using the conical approximation (14) (both applied element-wise to vectors). We ablate with approximations, which we call $l_1$, defined as

$$l_1(x, \alpha, \beta) = \|x - \alpha\|_1 + \|x - \beta\|_1, \tag{26}$$

and $l_2$, defined as,

$$l_2(x, \alpha, \beta) = \|x - \alpha\|_2^2 + \|x - \beta\|_2^2. \tag{27}$$

Notice that without any approximation, the indicator function has a zero gradient wherever it is differentiable, and hence gradient based optimization does not make any progress. Hence, we omit the option of no approximation from our ablation in Fig. 4.

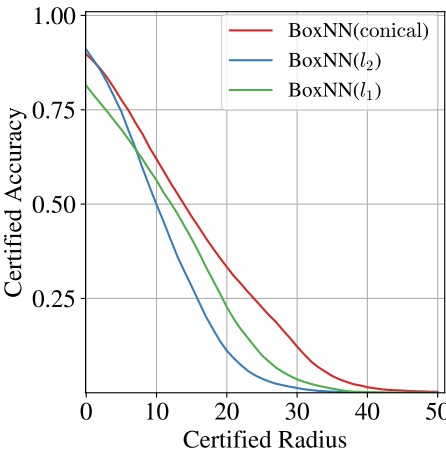

Figure 4: Ablation over the choice of the relaxation for the indicator function.

2. **Initialization.** For each box, we choose the location and size at initialization. For location, we find that centering boxes initially on training data points is always much better than random initialization, and hence we omit this from our ablation. We ablate on the size of the box $\alpha$, such that the boxes at initialization are defined as $\{(B(x - \alpha, x + \alpha), y) : (x, y) \in T\}$, for a subset $T$ of size $M$ chosen uniformly at random from the training data, as mentioned in Section 4.2. The resultant ablation is shown in Fig. 5.

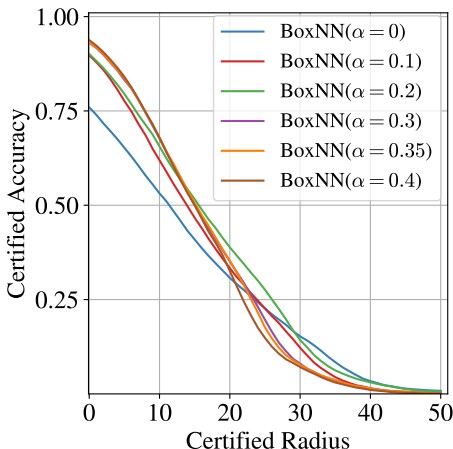

Figure 5: Ablation over the size of the inital boxes $\alpha$. Note that the initial $\alpha$ guides the optimization process to trade off benign accuracy (radius $= 0$) with certified accuracy at higher radii.

3. **Number of Boxes $M$.** The number of boxes $M$ above controls the expressive power of BOXNN. In order to maintain balance across classes, we assign an equal number of boxes for each of the 10 classes, and ablate on the number of boxes per class $M/10$ in Fig. 6.

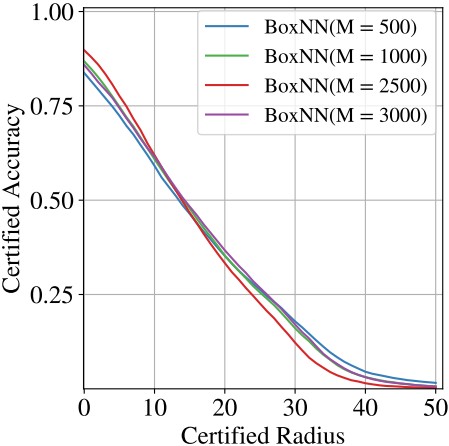

Figure 6: Ablation over the number of boxes $M$.

4. **Clipping Certificates** Recall that in (13), we clipped the robustness certificates in the optimization objective to $\beta = 50$ pixels to prevent a few boxes from dominating the loss. We ablate over this clipping value $\beta$ in Fig. 7.

5. **Optimizer.** We ablate over a few choices of the gradient-based optimizer for our problem: (a) vanilla SGD with a learning rate of 0.02, (b) SGD with a learning rate of 0.02, a momentum of 0.9, and a weight decay of 0.0005, and (c) Adam with a learning rate of 0.001, and standard decay factors, in Fig. 8.

6. **Approximation of** min. Recall that we replace the function min by an approximation to obtain non-sparse gradients to aid the optimization process. We describe the approximation $\min_\tau$ in Eq. (12), and we ablate over $\tau \in \{0, 0.5, 1, \infty\}$, where recall that $\min_0$ is same as the average function, and

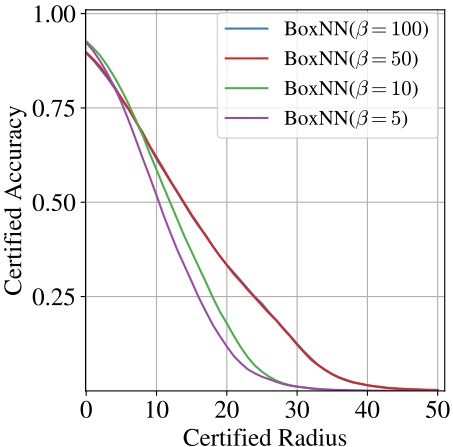

Figure 7: Ablation over certificate clipping value $\beta$. Note that the $\beta = 100$ line is almost the same as the $\beta = 50$ line, as there is no further effect due to clipping after 50.

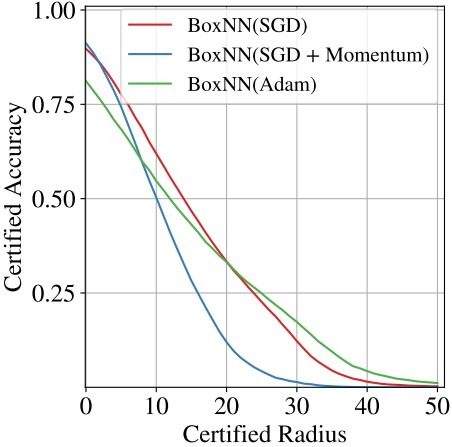

Figure 8: Ablation over the choice of the optimizer.

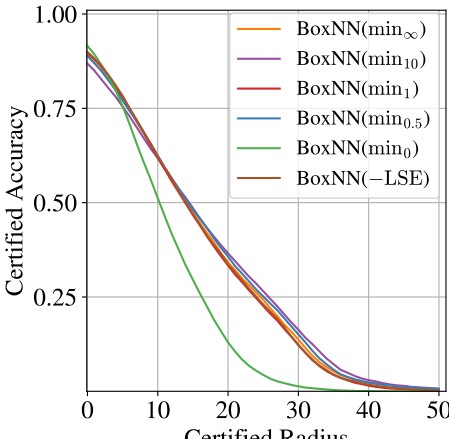

Figure 9: Ablation over Approximation of min.

that $\min_{\infty}$ is same as the standard min function. We also compare to an approximation by using the logsumexp function, as

$$\min(c_1, c_2, \ldots, c_M) \approx -\mathrm{logsumexp}(-c_1, -c_2, \ldots, -c_M) \overset{\mathrm{def}}{=} -\log \sum_{m=1}^{M} \exp(-c_m). \tag{28}$$

The resulting ablation is presented in Fig. 9.

### B.3 Additional Details and Certification Procedures

In this section, for the sake of completeness, we will briefly summarize the certification procedures developed in Levine & Feizi (2020a); Jia et al. (2022); Hammoudeh & Lowd (2023), which form our baselines for comparison in Section 5.

**Certification procedure for** RA **(Levine & Feizi, 2020a).** Recall from Eq. (15) that given a base classifier $f$, Levine & Feizi (2020a) produce a smoothed classifier $g$ as

$$\mathrm{RA}_\rho(x) = \arg\max_k \ \mathbb{P}_{v \sim \mathrm{Unif}(V_\rho)} \left( f(x \odot v) = k \right), \tag{(15) revisited}$$

where $V_\rho = \{v \in \{0,1\}^n : \|v\|_0 = \rho\}$ is the discrete set of all binary vectors of length $n$ having exactly $\rho$ ones, and $\odot$ denotes the Hadamard product. The method to produce a certificate of $\ell_0$ robustness in Levine & Feizi (2020a) follows the technique developed in Cohen et al. (2019), which is to bound the change in the probability assigned to the top class $k^\star$ in (15) when $x$ is perturbed. In other words, letting $p_k(x) = \mathbb{P}_{v \sim \mathrm{Unif}(V_\rho)} \left( f(x \odot v) = k \right)$ for $k \in \{1, 2, \ldots, K\}$, and $k^\star = \arg\max_k p_k(x)$, Levine & Feizi (2020a) show that

$$\text{if } p_{k^\star}(x) > 1.5 - \frac{\binom{n-r_{\mathrm{RA}_\rho}}{\rho}}{\binom{n}{\rho}}, \text{ then } \mathrm{RA}_\rho(x) = k^\star \ \forall x' \text{ such that } \|x' - x\|_0 \leq r_{\mathrm{RA}_\rho}. \tag{29}$$

Thus, the greatest quantity $r_{\mathrm{RA}_\rho}$ that satisfies the if condition in (29) becomes the $\ell_0$ robustness certificate for the classfier $\mathrm{RA}_\rho$ at the input point $x$.

However, obtaining of the probability $p_k(x)$ exactly is computationally infeasible, due to the exponentially large size of $V_\rho$, and hence evaluation of the classifier $\mathrm{RA}_\rho$ and the certified radius $r_{\mathrm{RA}_\rho}$ is computationally infeasible. Nevertheless, the technique to produce such randomized smoothing certificates in practice is to obtain a high confidence lower bound on $p_{k^\star}$ using several samples from $\mathrm{Unif}(V_\rho)$ following standard statistical estimation literature, and use this lower bound for computing $r_{\mathrm{RA}_\rho}$ (Cohen et al., 2019, Sec 3.2.2).

**Certification procedure for** $\mathrm{RA}^{\mathrm{B}}$ **(Jia et al., 2022).** Jia et al. (2022) produce a tighter analysis of the certificate of robustness for the classifier defined in (15). Their primary contribution is to develop techniques that can provide $\ell_0$ robustness certificates for the top-$k$ predictions where $k$ can be greater than 1. For the purposes of comparison to our work, we are only concerned for the special case of $k = 1$ in their certificate (Jia et al., 2022, Theorem 1). As Jia et al. (2022) remark, in this case, the form of the certificate is identical to Levine & Feizi (2020a), with the important difference that Jia et al. (2022) utilize the fact that each of the probabilities $p_k(x)$ are integer multiples of $\frac{1}{\binom{n}{\rho}}$, by incrementing the empirically computed lower bounds on these probabilities to the nearest integer multiple of $\frac{1}{\binom{n}{\rho}}$. As these modified probabilities $p'_k(x)$ satisfy $p'_k(x) \geq p_k(x)$, the largest radius $r_{\mathrm{RA}_\rho^{\mathrm{B}}}$ that satisfies the condition $p'_k(x) > 1.5 - \frac{\binom{n-r_{\mathrm{RA}_\rho^{\mathrm{B}}}}{\rho}}{\binom{n}{\rho}}$ is equal to or greater than $r_{\mathrm{RA}_\rho}$, resulting in a higher certified accuracy curve.

**Certification procedures for** FPA **(Hammoudeh & Lowd, 2023).** As opposed to the randomized defenses above, Hammoudeh & Lowd (2023) build a classifier that is simply a majority vote among several sub-classifiers, each looking at a subset of the pixels of the full input image. More formally, partitioning the set of pixels $\{1, 2, \ldots, n\}$ into disjoint partitions $\mathcal{V}$, recall that the classifier certified in Hammoudeh & Lowd (2023) is

$$\mathrm{FPA}^{\mathrm{A}}(x) = \arg\max_k |\{V \in \mathcal{V} : f(x_V) = k\}|, , \tag{(16) rewritten}$$

where $f(x_V)$ is the prediction obtained from a classifier that zeroes out the pixels of $x$ not in $V$. Defining the number of votes for class $k$ as $n_k = |\{V \in \mathcal{V} : f(x_V) = k\}|$ for $k \in \{1, 2, \ldots, K\}$, the winning class as $k^\star = \arg\max_k n_k$, and the runner-up class as $k^{\mathrm{ru}} = \arg\max_{k \neq k^\star} n_k$, Hammoudeh & Lowd (2023) present the robustness certificate by a simple counting:

$$\mathrm{FPA}^{\mathrm{A}}(x') = k^\star \ \forall x' \text{ such that } \|x' - x\|_0 \leq r_{\mathrm{FPA}^{\mathrm{A}}} \overset{\mathrm{def}}{=} \frac{n_{k^\star} - n_{k^{\mathrm{ru}}}}{2}. \tag{30}$$

For evaluating the above certificate $r_{\text{FPA}^{\text{A}}}$ for any input $x$, one needs to compute $n_k$ for all the classes. This is easily done by counting the votes for each class from each classifier $f(x_V)$, $V \in \mathcal{V}$.

Hammoudeh & Lowd (2023) then note that the above certificate discards the information from all the classifiers which did not predict $k^\star$ or $k^{\text{ru}}$, and remedy this in their improved certificate $r_{\text{FPA}^{\text{B}}}$. For this, the idea is to first determine $k^\star$ and $k^{\text{ru}}$, and then get *all* the classifiers $f(x_V)$ to vote for one of these two classes. The predicted label is the majority winner of this vote, *i.e.*,

$$\text{FPA}^{\text{B}}(x) = \text{Majority}\{\text{vote}_{f(x_V)}(k^\star, k^{\text{ru}})\}_{V \in \mathcal{V}}. \tag{31}$$

While the above description of a two-round voting is quite general, the actual voting scheme in Hammoudeh & Lowd (2023) is implemented by setting $f$ to be a neural network classifier, and the vote decided by which class among $\{k^\star, k^{\text{ru}}\}$ is assigned a large probability by the network. Evaluation of the certificate $r_{\text{FPA}^{\text{B}}}$ now becomes much more complicated due to the complicated relationship between the label predicted and $n_k$. Nevertheless, $r_{\text{FPA}^{\text{B}}}$ is typically larger than $r_{\text{FPA}^{\text{A}}}$, and we produce a comparison to our method BoxNN. Specifically, in Fig. 10, we compare BoxNN against the numbers reported in Hammoudeh & Lowd (2023, Table 27) ("FPA with run-off elections"). We observe that BoxNN improves upon the median certified robustness against FPA$^{\text{B}}$, and more generally mirrors the observations in Section 5 with respect to other baselines.

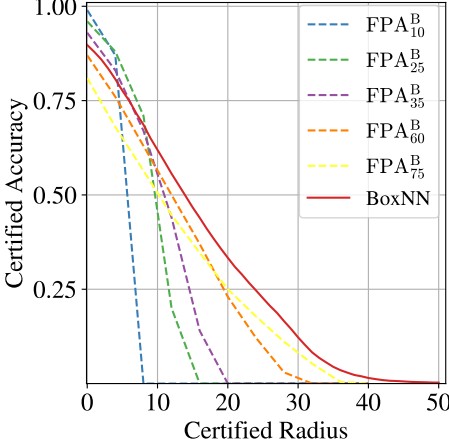

Figure 10: Comparison of Hammoudeh & Lowd (2023) (dotted lines) to our method Box-NN (red line) on the MNIST dataset. The dotted lines correspond to different settings for the hyperparameter $\rho$. Details are mentioned in Section 5.

