# OpenReview forum: "Certified Robustness against Sparse Adversarial Perturbations via Data Localization"
_TMLR — Accepted by TMLR_

### Review · Reviewer_HDkE · 2024-07-30

**Summary Of Contributions:**

This paper makes theoretical progress in the relationship between the classifier robustness against adversarial examples and the localization property of the data distribution. Specifically, the authors extend the existing work, which proved that the existence of a robust classifier against $\ell_2$-norm bounded attack implies the concentration property of the data distribution. As a result, the authors proved a similar implication: if there exists a robust classifier against $\ell_0$-norm bounded attack, the data distribution should be localized. Furthermore, the authors also proved that the converse is true with a stronger definition of localization. Then, the authors propose a new $\ell_0$-adversarially robust classifier, Box-NN. From a few experiments, the authors compare the performance of Box-NN to the existing methods with $\ell_0$-certifications.

**Audience:**

Yes

**Broader Impact Concerns:**

I don’t see a particular broader impact concern regarding this paper.

**Claims And Evidence:**

Yes

**Requested Changes:**

1. I’m interested in some practical applications of those theoretical results. If I understand correctly, Theorem 2.2 can be used to quantify the localization of the data distribution. Is it possible to quantify the localization of MNIST and Fashion MNIST data from the Box-NN classifier used in the experiment?
2. I’m not entirely convinced that the definition of localization needs to be compared to the traditional notion of measuring concentration. If this is not necessary, consider moving this discussion to the Appendix.
3. Can the experimental results for small certification radii (less than 5) be justified?
4. Rather than referring to papers to describe certification procedures, create a separate section in the Appendix and add a description of the experimental details.

**Strengths And Weaknesses:**

### Strengths
1. The authors make novel theoretical contributions to adversarial machine learning. Their modified definition of localization seems novel, and this notion will be useful for other researchers interested in the relationship between data distribution and the possibility of robust classification.
2. The experiments demonstrate that the proposed method has better-certified radii. This can be considered remarkable progress in the certified defense of adversarial examples.

### Weaknesses
1. As the authors pointed out, the proposed method is not easy to train efficiently.
2. It seems that Box-NN's experimental result for small certified radii is slightly worse than that of other methods.

---

> ### Author Response · Authors · 2024-09-05
> **Response to Reviewer HDkE**
>
> We thank the Reviewer HDkE for finding our work novel, and impactful to the progress in certified defenses. We address their comments in detail below:
>
> -------
>
> >  I’m interested in some practical applications of those theoretical results. If I understand correctly, Theorem 2.2 can be used to quantify the localization of the data distribution. Is it possible to quantify the localization of MNIST and Fashion MNIST data from the Box-NN classifier used in the experiment?
>
> Indeed, Thm 2.2 can be used to compute localization parameters of a data distribution. For MNIST and Fashion-MNIST, instead of having access to the full data generating distribution $p$, we only have access (via the corresponding test sets) of an empirical distribution $\hat p$, which assigns $1 / N_\text{test}$ probability mass to each data-point. For this distribution, we can empirically compute the $\delta$ parameter (i.e., $1 - \text{robust accuracy}$) corresponding to $\epsilon$ (i.e., number of pixels attacked) by directly reading off the red certified accuracy curve in Figure 2. Localization has one more parameter, $C$, which we still need to compute.
>
> To compute $C$, for all classes $k$, we need $\text{Vol}(A_k)$, i.e., the volume of the set $A_k$ where our classifier BoxNN predicts the label $k$. This is tricky to compute exactly, but note that any upper bound $C'$ on $C$, i.e., $C \leq C'$, suffices for our purposes (as the localization condition only requires an upper bound on $\text{Vol}(S)$). Now, from the last paragraph of Proof 2.2, we have $C = 1 / (1 - \text{Vol}(A_k))$, hence an upper bound on $C$ can be obtained from an upper bound on $\text{Vol}(A_k)$. For our classifier BoxNN, this can be obtained by summing the volumes of all the boxes belonging to classes other than $k$, and subtracting this quantity from $1 = {\rm Vol}(\mathcal{X})$.
>
> To summarize, using notation from Section 4.1, our empirical distribution is $(C', \epsilon, \delta)$-localized, where  $C' = 1 / \sum_{ \\{B_m \in \mathcal{B} \colon y_m \neq k\\} }\text{Vol}(B_m)$, and $(\epsilon, \delta)$ is any point on our certified accuracy curve. Notice that the above shows that, the same distribution admits a continuum of localization parameters depending on how much one is willing to trade-off $\epsilon$ for $\delta$, and is an important nuance of the definition of localization.
>
> --------
>
> >  I’m not entirely convinced that the definition of localization needs to be compared to the traditional notion of measuring concentration. If this is not necessary, consider moving this discussion to the Appendix.
>
> Thanks for bringing this up. We considered this change, and ultimately felt that the manuscript benefits from an explicit discussion of various nuances of this new notion of localization in light of the very well known notion of measure concentration, as both are quite similar in spirit. We will keep the discussion in the main text for now, but are happy to iterate further if needed.
>
> --------
>
> >  Can the experimental results for small certification radii (less than 5) be justified?
>
> Good question. Essentially, the baseline methods (Levine and Feizi, Hammoudeh and Lowd, Jia et al.) have underlying neural network based classifiers that are trained with an objective function aimed at increasing the benign accuracy. This ensures that the point on the certified accuracy curves corresponding to $\epsilon = 0$ is very high. On the other hand, our training objective aims to maximize the area under the certified accuracy curve over all $\epsilon \geq 0$ (see discussion before Eq. (11) on P9). This causes us to trade off the benign accuracy ($\epsilon = 0$) for a "higher" curve beyond $\epsilon = 5$. Even more, the number of parameters of BoxNN is proportional to $M$, the number of boxes we learn, and classification takes place in the input space. As such, BoxNN does not match the benign accuracy of the baseline methods which are based on highly overparameterized neural networks where classification takes place in a latent space specially learnt for the underlying task. Finally, as a part of our response to Reviewer RovY, we have now added an additional Appendix B.2 where we perform an ablation study on how the choice of parameters like $M$ affects our certified accuracy curve. In this ablation, we see that some of the parameters in BoxNN can indeed be modified to improve benign accuracy at the cost of a lower curve at higher radii.
>
> --------
>
> > Rather than referring to papers to describe certification procedures, create a separate section in the Appendix and add a description of the experimental details.
>
> Agreed. We have updated the PDF with this change by describing the certification procedures from the baseline methods (Levine and Feizi, Hammoudeh and Lowd, Jia et. al.) in Appendix B.3.
>
> The changes (individual sentences in the main text or section header in the Appendix) are colored teal in the updated PDF.

---

### Review · Reviewer_RovY · 2024-08-08

**Summary Of Contributions:**

The paper presents a technique for certification against sparse adversarial attacks. The paper shows that if a $l_0$-robust classifier exists for a data distribution, this distribution must be localized. Further, it shows that if the class conditional distributions are sufficiently separated, then it is sufficient for the existence of a robust classifier. Based on the theoretical results, the paper proposes a classifier robust against sparse adversarial attacks, called Box-NN. In the evaluation, Box-NN is shown to be better than prior works.

**Audience:**

Yes

**Claims And Evidence:**

Yes

**Requested Changes:**

See above

**Strengths And Weaknesses:**

Strengths:
- The paper presents novel theoretical results and shows it improves over the prior work $l_0$ classifier robustness
- The paper is well-written and gives good intuition on the theoretical results
- The paper has similarities to Pal et. al. 2023 work on $l_2$ robustness, and extends the theoretical results for  $l_0$ robustness. I appreciate the comparison and contrasting of the results of current paper and Pal et. al. 2023 throughout the paper.

 Weakness:
- The main weakness of the work is the challenge in learning classifiers that have axis-aligned decision regions. Thus, the experiments are limited to MNIST and Fashion-MNIST dataset. This limitation is clearly stated in the limitation section of the paper.


Comments:
> Contribution 1 and Def. 2.1

Define C here.

> The conditional distribution $pX|Y =k$ for each $k \in Y$ will be denoted by $q_k$.

Can you make a separate definition and discussion of class conditionals, given its importance throughout the paper

>  Definition 2.1

If $\epsilon$  is used as $l_0$ norm, can you use a different variable here for defining localized distribution $(C, \epsilon, \delta)$

> Theorem 2.2: “If the classes are balanced”

Can you formally state what balanced is?

> Theorem 2.2: regarding  $A_{k’} \subseteq \mathcal{X}$

What if $\mathcal{X} = A_{k’}$? What is C in this case?

>  Section 4.2 defines optimization tricks to maximize this lower bound

Can you do an ablation study to show the effect of these optimizations?

---

> ### Author Response · Authors · 2024-09-06
> **Response to Reviewer RovY**
>
> We thank the reviewer RovY for finding our results well written and novel, and for the careful reading of our manuscript. We address their comments below:
>
> -----
>
> > Ablation study to show the effect of training optimizations.
>
> We performed a detailed ablation study over all the components of our training, and created a new Appendix B.2 for these results. The ablation study shows how each training optimization affects the final certified accuracy curve. The ablation study also finds parameter settings for BoxNN which can be used to trade-off the benign accuracy at $\epsilon = 0$ for a lower certified accuracy at higher radii.
>
> -----
>
> > What happens when $A_{k'} = \mathcal{X}$ ? What is $C$ in this case?
>
> Great question! Thanks for catching this. When $A_{k'} = \mathcal{X}$, the classifier $f$ is constant - it predicts the class $k'$ over all of the input space. For most natural data distributions $p$, such a classifier $f$ would not obtain good robustness, i.e., the robust risk $R(f, \epsilon)$ would be close to $1$. However, if $p$ is degenerate enough so that $R(f, \epsilon)$ is close to $0$ even though $f$ is constant, the localization parameter $C$ guaranteed by Thm 2.2 becomes arbitrarily large, and the upper bound on the volume is not informative. This is expected, as the existence of a robust classifier on an almost degenerate distribution does not tell us much about the localization properties of the distribution.
>
> To demonstrate the above, we show that in this degenerate case, we can obtain the robustness parameters of $f$ exactly, without having any knowledge of the geometry of the class conditionals. Without loss of generality, let $k' = 0$, i.e., $f(x) = 0$ for all $x \in \mathcal{X}$. As in Proof 2.2, we can expand the robust risk as
> $R(f, \epsilon) = \sum_k q_k(U_k) \mathbb{P}(y = k)$. Now note that the unsafe set $U_0 = \\{ x \in \mathcal{X} \colon \exists \bar x \in B(x, \epsilon) \text{ such that } f(\bar x) \neq 0 \\}$ is empty as $f(\bar x) = 0$ for all $\bar x \in \mathcal{X}$. Similarly, $U_k = \mathcal{X}$ for all $k \neq 0$. Hence, the robust risk simplifies as $R(f, \epsilon) = \sum_{k \neq 0} \mathbb{P}(y = k) = 1 - \mathbb{P}(y = 0)$, showing that $f$ is exactly $(\epsilon, 1 - \mathbb{P}(y = 0))$ robust.
>
> We have updated the PDF to specify ``non-constant'' $f$ in the theorem statement, as the constant case is not interesting. Further, we are happy to add this discussion to the PDF.
>
>
> --------
>
>
> > Define and clarify $C$.
>
> $C$ is a parameter of the localization condition, just like $\epsilon$ and $\delta$. An explanation to this effect has added to the revised PDF.
>
> > Clarify class conditionals.
>
> Class conditionals are defined in the standard fashion, and an explanation has been added to the revised PDF.
>
> > Define balanced classes.
>
> A definition of balanced classes (i.e., the marginal distribution over the labels is uniform) has been added to the revised PDF.
>
> The changes (individual sentences in the main text or section header in the Appendix) are colored teal in the updated PDF.

---

### Review · Reviewer_yfCE · 2024-08-13

**Summary Of Contributions:**

The paper studies provable robustness of classifiers against $\ell_0$-bounded adversarial attacks. First, it provides theoretical results which connect the existence of $\ell_0$-robust classifiers and the localization property of the data distribution, building upon and extending the work of Pal et al. (2023). Moreover, the paper introduces Box-NN, a classifier which can provide by design certification on the $\ell_0$-robustness at little computational cost compared to existing methods. In the experiments on MNIST and F-MNIST, the paper shows how to efficiently train Box-NNs, and that these often outperform existing methods in terms of certified robustness.

**Audience:**

Yes

**Broader Impact Concerns:**

No concerns.

**Claims And Evidence:**

Yes

**Requested Changes:**

- [Minor] While the robustness curves provide a comprehensive picture, I think it'd be helpful to additionally have (in appendix) some tables with (a subset of) the corresponding results to e.g. easily read clean performance.

**Strengths And Weaknesses:**

Strengths:
- The theoretical results expand the existing works on understanding the existence of robust classifiers to a new threat model.

- Box-NNs are simple and can efficiently provide certified $\ell_0$-robustness. Moreover, in the experiments outperform existing methods especially at large radii.

- The paper is well-written, and the presentation of the different parts easy to follow.

Weaknesses:
- In practice, it seems that Box-NNs, given their simplicity, might struggle on more complex datasets e.g. of natural images like CIFAR-10. It seems, in fact, from Fig. 2 and Fig. 3 that already on MNIST clean performance is relatively low, and the drop would likely be larger compared to classifiers which don't aim at certified robustness.

---

> ### Author Response · Authors · 2024-09-05
> **Response to Reviewer yfCE**
>
> We thank Reviewer yfCE for finding our paper well written. We address their comment below:
>
> > While the robustness curves provide a comprehensive picture, I think it'd be helpful to additionally have (in appendix) some tables with (a subset of) the corresponding results to e.g. easily read clean performance.
>
> Agreed. We have added numerical details of the Figures to Appendix B.1 in the revised PDF, by sampling the $x$-axis of our Figures at equally spaced intervals and reporting the certified accuracies. We did the same for the baseline methods for an easy comparison.
>
> The changes (individual sentences in the main text or section header in the Appendix) are colored teal in the updated PDF.

---

### Decision · Action_Editor_H6cw · 2024-09-20

**Recommendation:** Accept as is

**Comment:**

The paper extends a recent theoretical result on certified adversarial robustness from l2 attacks to l0-bounded attacks, leading to a novel algorithm/architecture for neural classifiers that have certified robustness guarantees against l0-bounded attacks. The theoretical results are sound, non-trivial, and original; and empirical results validate the theoretical claims. Reviewers did not have much criticism in their initial review, other than empirical validation being limited to relatively simple datasets, with some theoretical concerns regarding the applicability of the method to complex datasets. The criticism has been sufficiently addressed by the authors, and all reviewers recommend accepting the paper. I agree with their recommendation, and since there are no open issues or remaining improvements, I argue for accepting the paper as is.

**Audience:**

The paper is interesting to the TMLR audience, particularly to researchers working on adversarial robustness and designing networks with theoretical robustness guarantees. All reviewers agree that this paper is interesting to the TMLR audience.

**Claims And Evidence:**

All reviewers agree that the claims made in the paper are supported by accurate, clear, and convincing evidence (already in their initial reviews, and no change to this after the rebuttal).